# HyperTraPS-CT: Inference and prediction for accumulation pathways with flexible data and model structures

**Olav N. L. Aga**[1,2], **Morten Brun**[3], **Kazeem A. Dauda**[3], **Ramon Diaz-Uriarte**[4,5], **Konstantinos Giannakis**[3,6], **Iain G. Johnston**[1,3]*

**1** Computational Biology Unit, University of Bergen, Bergen, Norway, **2** Department of Clinical Science, University of Bergen, Bergen, Norway, **3** Department of Mathematics, University of Bergen, Bergen, Norway, **4** Department of Biochemistry, School of Medicine, Universidad Autonoma de Madrid, Madrid, Spain, **5** Instituto de Investigaciones Biomedicas Sols-Morreale (IIBM), CSIC-UAM, Madrid, Spain, **6** Department of Disease Burden, Norwegian Institute of Public Health, Bergen, Norway

* iain.johnston@uib.no

**Data Availability Statement:** All code and data (with references) are available at https://github.com/StochasticBiology/hypertraps-ct.

## Abstract

Accumulation processes, where many potentially coupled features are acquired over time, occur throughout the sciences from evolutionary biology to disease progression, and particularly in the study of cancer progression. Existing methods for learning the dynamics of such systems typically assume limited (often pairwise) relationships between feature subsets, cross-sectional or untimed observations, small feature sets, or discrete orderings of events. Here we introduce HyperTraPS-CT (Hypercubic Transition Path Sampling in Continuous Time) to compute posterior distributions on continuous-time dynamics of many, arbitrarily coupled, traits in unrestricted state spaces, accounting for uncertainty in observations and their timings. We demonstrate the capacity of HyperTraPS-CT to deal with cross-sectional, longitudinal, and phylogenetic data, which may have no, uncertain, or precisely specified sampling times. HyperTraPS-CT allows positive and negative interactions between arbitrary subsets of features (not limited to pairwise interactions), supporting Bayesian and maximum-likelihood inference approaches to identify these interactions, consequent pathways, and predictions of future and unobserved features. We also introduce a range of visualisations for the inferred outputs of these processes and demonstrate model selection and regularisation for feature interactions. We apply this approach to case studies on the accumulation of mutations in cancer progression and the acquisition of anti-microbial resistance genes in tuberculosis, demonstrating its flexibility and capacity to produce predictions aligned with applied priorities.

## Author summary

Many important processes in biology and medicine involve a progressive buildup of features over time. These might be, for example, the accumulation of different mutations as cancer progresses, or the evolution of bacteria to be resistant to more and more drugs.

**Funding:** This project has received funding from the European Research Council (ERC) under the European Union's Horizon 2020 research and innovation programme (Grant agreement No. 805046 (EvoConBiO) to IGJ). This work was supported by the Trond Mohn Foundation [project HyperEvol under grant agreement No. TMS2021TMT09 to IGJ], through the Centre for Antimicrobial Resistance in Western Norway (CAMRIA) [TMS2020TMT11]. RDU was partially supported by grant PID2019-111256RB-I00 funded by MCIN/AEI/10.13039/501100011033. The funders had no role in study design, data collection and analysis, decision to publish, or preparation of the manuscript.

**Competing interests:** The authors have declared that no competing interests exist.

Here we introduce an algorithm called HyperTraPS-CT that uses data to learn the details of how these features build up over time. The algorithm provides information on which features affect each other, which come early and which come late, and what might happen in the future. It is more flexible than several existing approaches, and can be used across many different scentific situations; we demonstrate its use in learning about leukemia progression and tuberculosis drug resistance. This approach has the potential to help make useful predictions about how new instances of these processes will evolve, about data which can't be observed due to technological limitations, and about possible mechanisms that determine how features interact.

## Introduction

Many important processes in the biological, medical, and physical sciences can be classed as 'accumulation processes' [1, 2], involving the serial stochastic acquisition or loss of discrete features over time—from evolutionary dynamics [3–5] to disease progression [6–8]. Features (also called traits, or characters, particularly in the evolutionary literature) in this context typically mark the presence or absence of a property of interest—for example, a particular mutation in a cancer patient [2, 6], a given gene lost by a species [4], or a given disease symptom presented by a patient [9]. Reconstructing the dynamics by which these processes occur can inform our knowledge of the underlying mechanisms [4, 5], make predictions about unmeasured features and the likely future behaviour of systems in a known state [5, 8, 10–12], and identify features of the system which determine (or are determined by) progress through accumulation pathways [9, 13].

The study of the evolution of characters across phylogenetically related lineages has an extensive history and associated literature, but general approaches suitable for large sets of data and features remain challenging. Methods to infer phylogenetic trees, to infer evolutionary dynamics of characters on phylogenies, and to jointly infer both have been developed (reviewed, for example, in [14] and [3] and included in famous software packages like *phytools* [15] and *corHMM* [16]). Another branch of the scientific literature, which remains surprisingly disconnected from the evolutionary picture, focusses on inferring accumulation dynamics in cancer progression (recently reviewed in [2] and [6]; a summary of some approaches in given in Table A in S1 Appendix). These approaches attempt to describe the acquisition of coupled features—usually mutations—with time, with classic approaches including Markov modelling of transitions between discrete states with disease observations [17]. Inference methods based on Bayesian networks have played a particularly pronounced role in this field [7, 18–23]. Traditionally (but not exclusively), the cancer literature considered independent, cross-sectional observations of a limited number of binary features, constraining the possible interactions between features to, for example, pairwise positive influences (but see below).

In an attempt to relax these restrictions, and to support cross-sectional, phylogenetic, and/ or longitudinal observations, 'hypercubic transition path sampling' or HyperTraPS was developed [4, 24]. HyperTraPS requires no assumptions about restricted states or independence of feature acquisitions, and is naturally embedded in a Bayesian framework supporting prior information and uncertainty quantification. A focus on transitions between states as the fundamental observation type, rather than individual observations, means that HyperTraPS has always supported data embedded in trees and/or longitudinal sequences, as well as cross-sectional samples. Polynomial rather than exponential scaling in the number of features, and at most linear scaling in the number of observations, permits the efficient analysis of large sets of

coupled features where other approaches struggle [4, 24]. This efficiency arises from a key property of the approach: an algorithm conceptually similar to transition interface sampling or forward flux sampling in statistical physics [25], focussing only on those pathways likely to correspond to observed transitions. While pairwise positive and negative interactions (involving $L^2$ parameters) are the default target of inference in HyperTraPS, regularisation and model selection approaches support a choice between different parameterisation structures for a given dataset [24]. Prior to and following HyperTraPS, two related approaches—phenotypic landscape inference [5] and HyperHMM (hypercubic hidden Markov modelling, [26])—expanded the parameter space of accumulation models to allow arbitrary interactions between sets of features (not just pairwise interactions), so that, for example, a combination of traits *A*, *B* can influence trait *C* differently from the additive influence of *A* and *B*. HyperTraPS and these aligned approaches have been used to explore cancer progression, identifying new pathways and interactions [24, 26], but their flexibility has also allowed their application in other fields including the evolution of genomes [4, 27]; multi-drug resistance in tuberculosis [24, 28]; photosynthetic pathways [5, 29], and tool-use behaviour [13]; disease progression in severe malaria [9]; and the behaviour of students in online learning [30]. Predictions from these inferred models about unobserved features and future behaviours have been validated both using withheld data [9] and independent laboratory experiments [5]. HyperTraPS is certainly not alone in this breadth of application; other approaches from accumulation modelling have also been expanded into other applied fields, notably in the study of HIV drug resistance [31, 32].

Independently of HyperTraPS, Mutual Hazard Networks (MHN) has been developed more recently [6, 33], driving the cancer accumulation field forward—along with other powerful advances including those based on Bayesian networks [1, 2, 34, 35], permutation analysis [36, 37], and accounting for tumour 'phylogenetics' (reviewed in [38]). MHN uses the same principle as the pairwise, $L^2$, parameterisation of HyperTraPS to support pairwise positive and negative influences between traits (but does not support higher-order interactions). MHN has recently been embedded in a framework, TreeMHN, allowing observations to be connected in a tree [12], matching the native capacity of HyperTraPS to deal with phylogenetically embedded and longitudinal data. To deal with larger sets of features, TreeMHN introduced a sampling approach to the MHN picture, aligning with HyperTraPS' sampling system.

One difference between these connected approaches is their picture of time. MHN and TreeMHN, for example, contain an implicit sampling time that sets a continuous timescale for the inferred dynamics; this feature has been a target of refinement and subsequent computational acceleration in [39]. HyperTraPS and HyperHMM consider only orderings, not timings, of transitions—originally motivated by the absence of timing information in the evolutionary and disease progression settings under study. In many evolutionary applications (and now cancer progression) of accumulation modelling, however, imperfect timing information does exist, in the form of (uncertain) branch lengths in the associated trees—either corresponding to real time or to amounts of evolutionary change. The capacity to deal with uncertain timings is also useful in cancer and disease progression, where observations are typically bounded by inequalities: changes occur at some unknown time between two delineating observation times.

Here, we develop HyperTraPS-CT, an expansion and generalisation of HyperTraPS that connects with absolute timings, either specified precisely or via a range of possible values, allowing uncertainty in observation times to be naturally included. HyperTraPS-CT retains all the existing strengths of HyperTraPS: scalability (having been used for over 120 pairwise-coupled features in [30]); flexibility in data source (cross-sectional, longitudinal, phylogenetic); the capacity for regularisation and model selection [24]; and a Bayesian implementation. New components of HyperTraPS-CT include the ability to use precise and/or uncertain timing

information; the ability to capture arbitrary positive and negative interactions between sets of (not just pairwise) features (mirroring HyperHMM in [26]); and an expanded set of options for predicting unobserved features in observations and future behaviours. We also introduce a suite of visualisation approaches reporting the output of these inference processes, which can readily be applied to other continuous-time models like MHN [33], CBN [22], and H-ESBCN [35], with a flexible implementation (via command line and R) allowing efficient use of these advances across platforms.

## Methods

### HyperTraPS-CT: Inferring timed evolutionary pathways on a hypercubic transition network

We will work in a picture where observed states of a system are described by the presence or absence of $L$ binary traits. There are in total $2^L$ different states (that is, unique patterns of presence or absence) which we will refer to with binary strings of length $L$, with 0 or 1 in the $i$th position respectively corresponding to the absence or presence of the $i$th trait (Fig 1A). We allow Poissonian transitions between states of the system with a characteristic rate: the rate of a transition from $s_1$ to $s_2$ is $\lambda_{s_1 \to s_2}$. In monotonic accumulation, where dynamics proceed by individual changes of one trait at a time, $\lambda_{s_1 \to s_2} = 0$ for all $s_2$ that do not differ from $s_1$ by an acquisition of exactly one trait. For example, for $L = 3$, $\lambda_{000 \to 001}$ may be nonzero, but $\lambda_{000 \to 011} = 0$.

Several early instances of HyperTraPS were designed to study systems where *loss* of traits, rather than accumulation, was the driving dynamic (for example, the loss of genes in the reductive evolution of mitochondrial DNA [4]). HyperTraPS can readily describe loss dynamics as well as accumulation dynamics, in which case the description above is inverted: $\lambda_{s_1 \to s_2} = 0$ for all $s_2$ that do not differ from $s_1$ by a *loss* of exactly one trait. In either case, the network of transitions between states takes a hypercubic structure (Fig 1B). Evolutionary trajectories of the system are modelled as random walks from some initial state, undergoing transitions randomly according to the rates of available transitions from the current state.

We consider datasets of the form $\mathcal{D} = \{a_i, b_i, \tau_{1i}, \tau_{2i}\}$, consisting of a collection of $N$ records of ancestral state $a_i$, descendant state $b_i$, and a observation time window ($\tau_{1i}, \tau_{2i}$) (Fig 1A). This window can be used to describe specified, uncertain, or relaxed constraints on observation times (see below). To compute the likelihood of an observation in our dataset, and thus make progress inferring the transition rates that are compatible with observations, we require the probability $P(b, \tau_1, \tau_2 | a, 0; \lambda)$ that, if a system is in (ancestral) state $a$ at time $t = 0$, it will be in (descendant) state $b$ at some time between $t = \tau_1$ and $t = \tau_2$, given a particular set of transition rates $\lambda$. If it is computationally feasible to analyse all the paths leading from $a$ to $b$, this probability can be computed, similar to MHN [33] (see S1 Appendix). However, if we are working with many traits, the number of paths leading from $a$ to $b$ may be computationally unreasonable to fully sample—particularly if this calculation is inside a loop, for example in a Bayesian search over parameter space. In such cases, we need to employ a sampling scheme that captures the pathways that are most likely. We first define a state $s$ as $b$-compatible if $s$ has acquired no features that $b$ does not have (hence, $b_i = 1$ for every $i$ where $s_i = 1$; this definition is inverted for loss dynamics). For example, in the case of feature acquisition, 011 is 001-compatible but not 100-compatible. The HyperTraPS algorithm [4] gives us a way of constructing paths starting at $a$ that are guaranteed to be $b$-compatible, and thus to end at $b$, allowing us to avoid wasting computational time analysing paths that will not correspond to observations.

In Algorithm 1 we present an approach to estimate $P(b, \tau_1, \tau_2 | a, 0; \lambda)$, which we call Hyper-TraPS-CT (hypercubic transition path sampling in continuous time). Eq 1 gives the central,

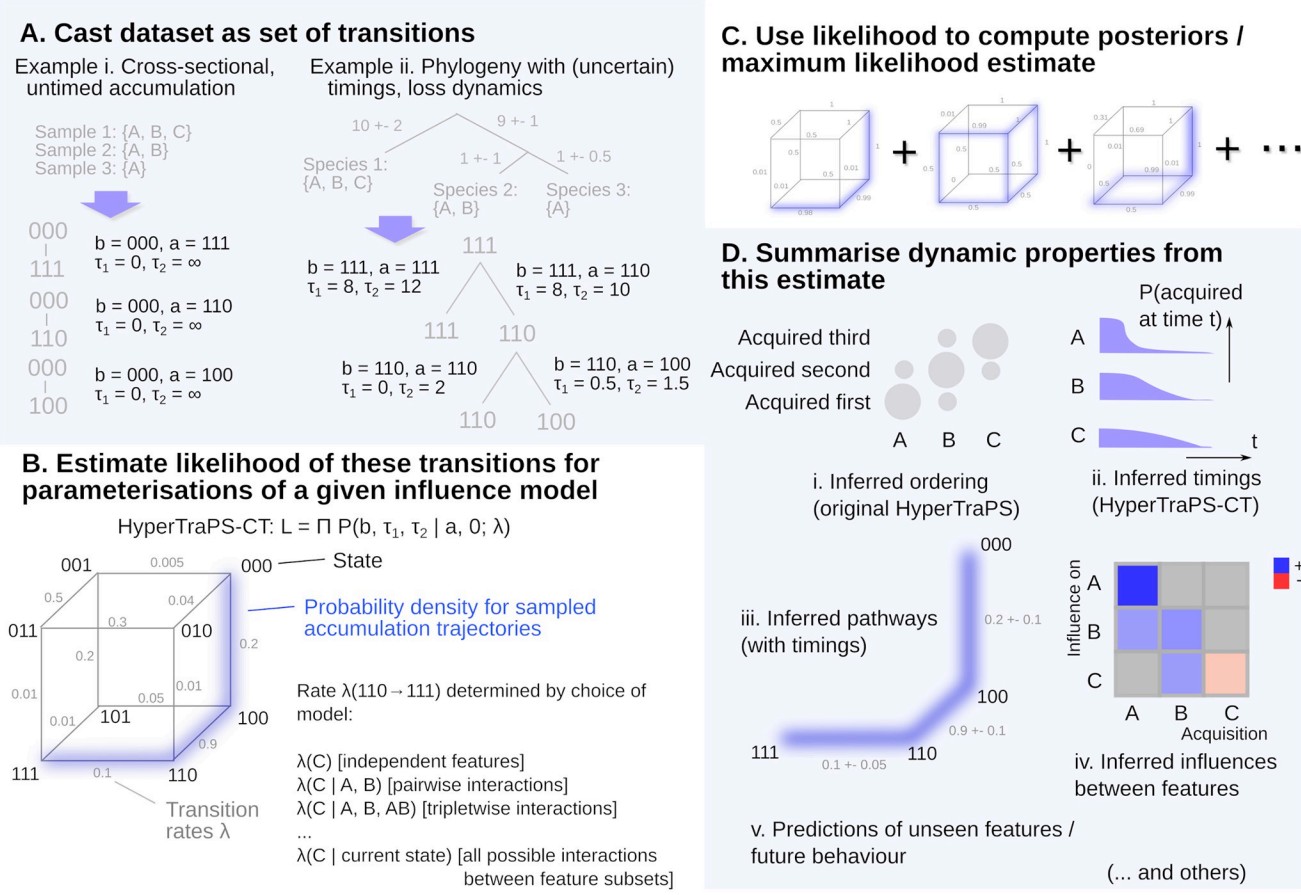

**Fig 1. Outline of HyperTraPS-CT approach. (A)** The relationship between observations is used to create a set of observed transitions. If observations are independent (i)—for example, individual patients, or independent lineages—they can be treated individually by imposing an initial state. If they are longitudinally or phylogenetically related (ii), transitions are inferred from the phylogeny with associated precise timings or uncertain time windows, or with completely unspecified timings where no timing information is available. **(B)** A hypercubic transition network is used to describe evolutionary pathways; Eq 1 can be used to estimate, using sampling, a likelihood of observations given a parameterisation of this network λ. **(C)** Different parameterisations are explored; for a Bayesian analysis, MCMC and any prior information is used to sample from posterior parameterisations that have a high associated likelihood given the data; for a maximum likelihood estimate the likelihood is optimised. **(D)** The identified parameterisation(s) can then be used to report orderings (i), timescales (ii), pathway structures (iii), influences between features (iv) that are likely given the set of observations, and to make predictions of unseen features and/or future behaviour (v).

general expression for the probability estimate (denoted $\hat{P}$). Some special cases merit further attention; we will treat the λ dependence as implicit for these. First, if $a$ and $b$ are identical, the expression reduces to the exact form $P(b, \tau_1, \tau_2 | b, 0) = \exp(-\beta_b \tau_1)$ (Step 1 in Algorithm 1), simply describing the probability that the system is still in state $b$ after a time window $\tau_1$. Second, if $\tau_1 = 0$ and $\tau_2 = \infty$, we obtain the probability that, given that the system is in state $a$ at $t = 0$, $b$ is encountered at *any* time in the future. Eq 1 then reduces to $\hat{P}(b, 0, \infty | a, 0) = \frac{1}{N_h} \sum_{c \in C} \alpha(c)$, summing properties $\alpha(c)$ of path $c$ over the set of paths $C$ sampled by $N_h$ independent random walkers (in HyperTraPS(-CT), these random walkers are constrained to follow only $b$-compatible paths—that is, paths that will contribute to the likelihood calculation—and the amount of constraint this entails is recorded and used in the likelihood estimation). This is exactly the quantity reported by the original HyperTraPS algorithm, without considering continuous time. Third, if $\tau_1 = \tau_2 = \tau$, we enforce that the system must be in state $b$ at an exact time $\tau$ after it is observed in state $a$, corresponding to an exactly-specified time window between the two observations.

**Algorithm 1.** Hypercubic transition path sampling in continuous time (HyperTraPS-CT). Requires a start state $a$, end state $b$, time window $\tau_1, \tau_2$, and transition rates $\lambda$.

1. Compute escape rate from $b$, $\beta_b = \sum_s \lambda_{b \to s}$. If $a \equiv b$, compute $P(b, \tau_1, \tau_2 | b, 0) = \exp(-\beta_b \tau_1)$ and terminate.
2. Initialise a set $C$ of $N_h$ trajectories at $a$.
3. For each trajectory $c$ in $C$:
   (a) Compute the probability of making a move to a $b$-compatible next step; record this probability as $\alpha'^{(c)}$.
   (b) Record the sum of rates of processes escaping from the current state: $\beta_{c_i} = \sum_s \lambda_{c_i \to s}$.
   (c) If current state is $a$, set $\alpha^{(c)} \to \alpha'^{(c)}$, otherwise update $\alpha^{(c)} \to \alpha^{(c)} \alpha'^{(c)}$.
   (d) Select one of the available $b$-compatible steps according to their relative weight. Update trajectory ($c_i \to c_{i+1}$) by making this move.
4. If current state (in all trajectories) is $b$ go to 5, otherwise go to 3.
5. Record $\alpha(c) = \alpha^{(c)}$ for each path $c$. As in original HyperTraPS, $\hat{P}(a \to b) = N_h^{-1} \sum_c \alpha(c)$. Use recorded $\{\beta_{c_i}\}$ to compute $v_i(c) = (\prod_{j=1}^{n-1} \beta_{c_j}) / \prod_{j=1, j \neq i}^{n-1} (\beta_{c_j} - \beta_{c_i})$.
6. Compute

$$\hat{P}(b, \tau_1, \tau_2 | a, 0) = \frac{1}{N_h} \sum_{c \in C} \alpha(c) \sum_i \left( e^{-\beta_b \tau_1} \frac{v_i(c)(1 - e^{-\tau_1(\beta_{c_i} - \beta_b)})}{\beta_{c_i} - \beta_b} + \frac{v_i(c)}{\beta_{c_i}} \left( e^{-\beta_{c_i} \tau_1} - e^{-\beta_{c_i} \tau_2} \right) \right), (1)$$

The derivation of Eq 1, Algorithm 1, and these specific results is provided in S1 Appendix, along with further mathematical details. Briefly, we exploit two facts from the assumption of Poissonian dynamics. First, the 'arrival time'—the sum of the transition times through the pathway $c$—follows a hypoexponential distribution, which can be integrated over all times in the window from $\tau_1$ to $\tau_2$. Second, the 'dwell time'—the time the system remains at $b$—follows a simple exponential form; accounting for all patterns of arrival and dwell times then characterises the probability of a given observation.

Given the estimated probability from Algorithm 1, we construct an approximate likelihood associated with the dataset $\mathcal{D}$: $\mathcal{L}(\mathcal{D}|\lambda) \simeq \prod_i \hat{P}(b_i, \tau_{1i}, \tau_{2i} | a_i, 0; \lambda)$. This likelihood function now enables us to perform inference of the transition networks $\lambda$ most compatible with observed data (Fig 1C). In S1 Appendix we present a Bayesian MCMC algorithm that produces posterior distributions on transition rates given observations (Fig 1C and 1D) and naturally allows prior information on the system to be included in the analysis. We also use a variety of test cases to demonstrate that this approach can infer the true parameterisations of synthetic evolutionary state spaces, and can accurately reconstruct the orderings and timescales of evolutionary events. In addition, we demonstrate how the inclusion of prior information about the evolving system (for example, forbidding some transitions) can be used to increase efficiency and refine the resultant posteriors. All code for these test cases is available at https://github.com/stochasticbiology/hypertraps-ct, with illustrative examples at https://github.com/StochasticBiology/hypertraps-ct/blob/main/docs/hypertraps-demos.pdf.

In Figs A-B in S1 Appendix we demonstrate a set of calculations and verification case studies for the principle of HyperTraPS-CT. The distribution of inferred timescales for an example set of transitions matches the analytic result for Poisson dynamics under different parameterisations (Fig B in S1 Appendix). The rates on transitions for a model involving a single, simple pathway and a random set of transition rates across multiple pathways are well captured in the inference process, with strong discrepancies only arising for a limited set of rare transitions (Fig B in S1 Appendix). The time scaling of the likelihood estimation step is $\mathcal{O}(nL^2 N_h)$, where $n$ is number of observations [4]. Embedded in an inference process, $N_h$ (controlling the

stability of the likelihood estimate) and the number of likelihood calculations required may vary according to convergence criteria, as we explore below.

## Results

### Basic outputs from HyperTraPS-CT

Fig 2 gives a first example of output from HyperTraPS-CT, initially using the $L^2$ parameterisation as also used in mutual hazard networks (MHN, [33]). Here, the source data is generated from a process supporting two competing accumulation pathways (as used in [24] and [26]). One pathway involves progressive accumulation of feature 1, then 2, then 3, and so on. The other pathway involves progressive accumulation of feature $L$, then $L − 1$, then $L − 2$, and so on. Each progressive accumulation step takes 0.1 time units to occur.

Fig 2 demonstrates some outputs from inference for the $L = 5$ case. Fig 2A shows transitions through the hypercubic state space inferred to occur with high probability, and their inferred associated timescales. Fig 2B and 2C show maps combining the base rates of acquisition of each feature, and how the acquisition of each feature influences the rate of each other feature. Fig 2B follows the protocol of [12], representing the $L^2$ model parameters as a matrix. The

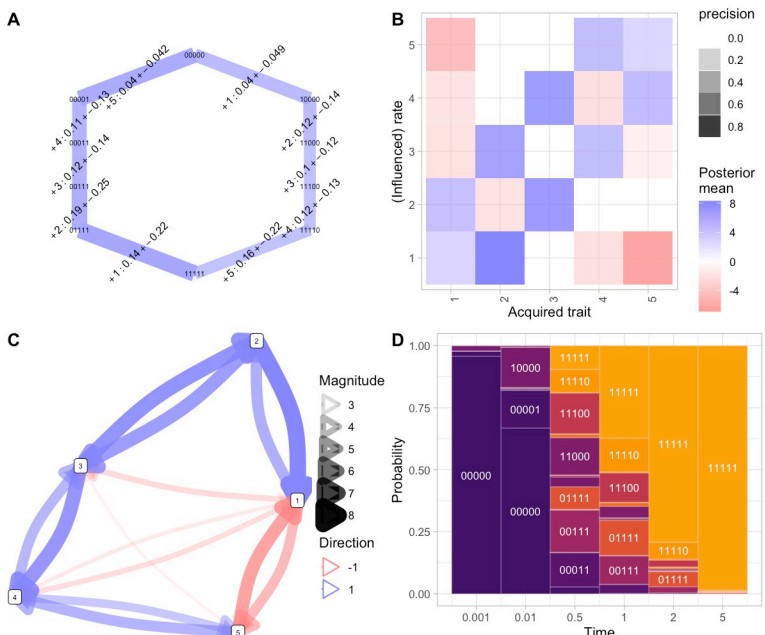

**Fig 2. Basic HyperTraPS-CT output for competing pathways.** *Italics* refer to plot type from Table 1, where more detail can be found for each. (**A**) *Hypercubic transition network* giving most probable inferred transitions, demonstrating the two competing pathways (left, features 1, 2, 3, . . .; right, features 5, 4, 3, . . .). Widths of edges give the probability flux through each edge; edge labels give the feature acquired and the range of associated timings for that transition; node labels give states. (**B**) *Influence matrix* following the plot styling in [12], summary of posterior distributions on the rates of the $L^2$ parameterisation for this system. Diagonal elements give the base acquisition rates for each feature; off-diagonal elements give the influence of an acquired feature (column) on the acquisition of another feature (row). The opacity of each point reflects its posterior width: 'precision' here is max(0, $1 − CV$) where $CV$ is the posterior coefficient of variation. The cross-repression of the first two steps, and promotion of the consequent pathway steps, is clear. (**C**) *Influence graph* of inferred influences between features, reporting the elements of (B) filtered for low CVs (hence with high posterior probability of being nonzero). (**D**) *State probabilities with time*: probabilities of different states (heights of rectangular regions) at different snapshot times during system evolution (horizontal axis).

Bayesian implementation of HyperTraPS-CT allows uncertainty to be quantified: here, a parameter's 'precision' is reported as $\max(0, 1 − CV)$ using the coefficient of variation of that parameter's posterior. Fig 2C provides a network representation mirroring Fig 2B, which we can later generalise to the case where collections of features influence each other. Fig 2D gives predicted states of the system at a collection of 'snapshot' observation times. Throughout these plots, the two-pathway structure is clear, including in the specific pathways in Fig 2, the patterns of base rates and cross-repression between features in Fig 2B and 2C, and the predicted states of the system in Fig 2D.

## Flexible inference with different data and model structures, optimisation, regularisation, and uncertainty

**Cross-sectional, longitudinal, and phylogenetic datasets.** HyperTraPS-CT can naturally handle cross-sectional data (imposing a given 'ancestral state' $a_i$ for each $b_i$ observation, for example $a_i = 0^L$ in the case of accumulation dynamics starting from an initial state with no traits acquired), longitudinal data (decomposed into $(a_i, b_i)$ pairs through the time course, independent by the Markov property), and phylogenetically embedded data ($a_i$ corresponding to ancestral nodes and $b_i$ to descendant nodes, with each $(a_i, b_i)$ pair again Markov independent). Fig 1 illustrates these cases. In this final case, a method for reconstructing ancestral states from modern observations is typically required. If feature acquisitions are assumed to be rare and irreversible events, this process is normally straightforward for accumulation processes: we assume that an ancestor had acquired a feature if all its descendants have it, otherwise we assume that the ancestor did not have the feature and any descendants possessing it acquired it independently (the ancestral state is given by the bitwise AND operator applied over descendants). This picture is readily inverted in the case of loss dynamics, where the bitwise OR operator would apply. However, these simple rule-based pictures will undercount instances of convergent evolution—where two descendant lineages independently acquire (or lose) a feature relative to their common ancestor. For more complex dynamics, tools from phylogenetic reconstruction like the Camin-Sokal [40] or Wagner [41, 42] methods can be used. In all cases, the $(a, b)$ form for an observation—with an associated time window, which may be infinite, wide, or precise (zero-width)—unifies the input data structure.

**Parameter structures: Zero, pairwise, setwise, or arbitrary interactions between features.** HyperTraPS-CT accepts a range of different parameter structures. The first trivial case, only of applied use as a null hypothesis, is the zero-parameter case, where every feature is acquired independently with the same rate. The next case (independent features) involves a parameter for the 'base rate' of acquisition of each feature $F_i$, for a total of $L$ parameters. The next case (pairwise interactions) involves these base rates and $L^2 − L$ further interaction parameters, describing how the acquisition of feature $F_i$ influences the base rate of feature $F_j$. With a total of $L^2$ parameters, this setup was introduced by [17], and generalised to allow negative interactions in HyperTraPS [4] and published independently as mutual hazard networks [33]. Extending those methods, we continue here, with an '$L^3$' parameterisation allowing pairs of features $F_i$, $F_j$ to have non-additive effects on the acquisition of feature $F_k$; an '$L^4$' parameterisation allowing triples $F_i$, $F_j$, $F_k$ to have effects independent from their constituent pairs; higher-order models can in principle also be applied. The limiting case is where each of the $L2^{L−1}$ edges on the hypercube have independent rates, used in HyperHMM [26] and corresponding to the case where subsets of features of all possible sizes can have independent influences on a transition.

HyperTraPS-CT allows inference using any of these model choices, with different capacities for interactions between features. The case where every edge on the transition network has an independent weight is likely to correspond to overfitting for reasonable cases. Here, individual parameter values will not all be uniquely identifiable or informative, many different parameterisations may give the same likelihood, and the task more resembles machine learning (finding a parameterisation that generates useful predictions) than parameter inference (identifying interpretable values for given parameters). Conversely, assigning parameters based on individual features alone is likely to underfit data generated by processes where interactions between features are important. Here, parameters will certainly take interpretable values, but may omit important mechanistic information about interactions. The $L^2$ case risks omitting information about triplet interactions, the $L^3$ case about quartets, and so on. Model selection and regularisation (described below) can be used to find the optimal parameter structure for the details of a given dataset.

In Fig 3A we give an example of how different parameter structures omit or capture different mechanisms. The data in this example are generated from a process where pairs exert different influence on feature acquisitions than their constituent individual members. Specifically, the accumulation pathway of the final three features is determined by whether two, or three, of the first three features have been acquired.

As in [26], the pairwise-interaction picture ($L^2$ here and in [24]; also mutual hazard networks [6, 12, 33]) cannot capture these higher-order interactions. Comparing Fig 3Ai with 3Aii, the $L^2$ parameterisation is forced to assign nonzero probabilities to a range of pathways that are not present in the generating model, because of the requirements of adjusting lower-order rate parameters to estimate the influence of higher-order processes. The $L^3$ parameterisation (Fig 3Aiii) has parameters supporting the nonadditive influence of pairs of features on acquisition rates—as in the generating model—and hence captures the dynamic structure. The corresponding inferred network of interactions, generalising the matrix picture of Fig 2B, is shown in Fig 3Av. The all-edges model in Fig 3Ai is comparable to the target of inference in HyperHMM [26]; as such a highly-parameterised model will typically reflect overfitting, regularisation can be used to prune extraneous parameters, leaving only those higher-order interactions needed to describe the data (Fig 3Aiv and 3Avi; see next subsection).

**Model regularisation.**   In addition to specifying a basic model structure from one of these families, HyperTraPS-CT supports variable selection through regularisation. This can be performed in several ways. First, stepwise parameter removal (as in [24]) where, after a model is fitted, the parameters are progressively set to zero, with the parameter removed at each step being the one that has least influence over the likelihood at that step. The minimum AIC (or other criterion) parameterisation can then be chosen, retaining the set of interactions that are necessary and sufficient to best describe the data (Fig 3Avi). Second, using a penalised likelihood, where model complexity is included as a penalty in either the maximum-likelihood optimisation or the MCMC process (see below). The penalised complexity can either be the number of nonzero parameters, following an information-criterion-like approach, or the magnitude of parameter values, following a LASSO-like approach. This approach is used in the tuberculosis and cancer case studies below (Figs 4 and 5). We generally found penalising the number of nonzero parameters (akin to an AIC-like approach) to give reproducible and robust results, but the best approach (and whether to penalise at all) will in general depend on the scientific question.

**Dataset size.**   HyperTraPS-CT's sampling approach means that it does not suffer a combinatorial explosion of paths that must be considered as $L$ increases. Although performing adequate sampling to characterise large systems is still a challenge, it is not an insurmountable one for some examples of reasonable size. HyperTraPS has been used successfully for $> 120$

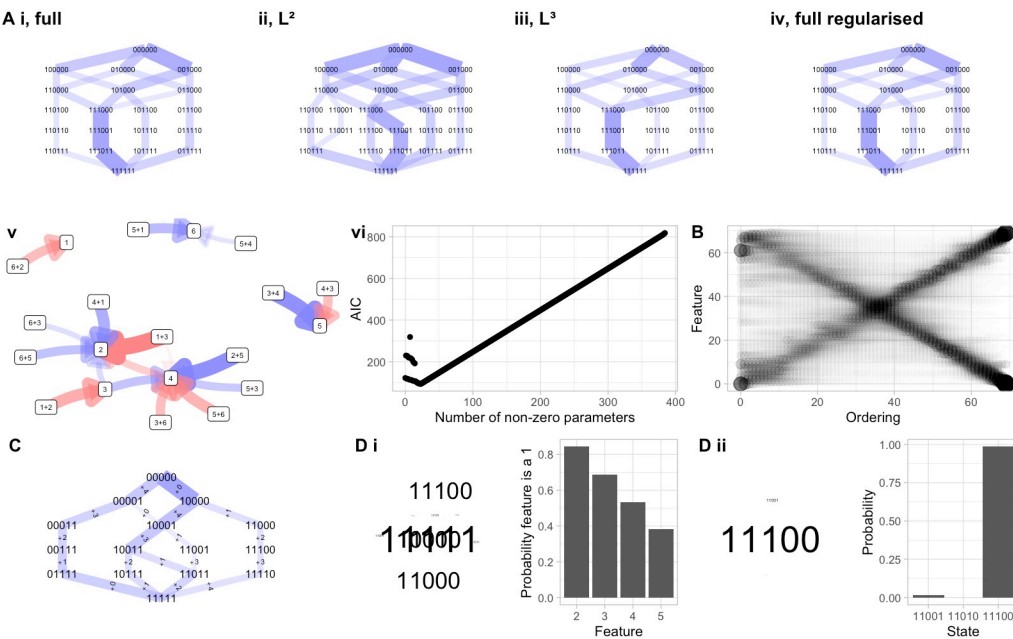

**Fig 3. Model structure, scaling, priors, and predictions with HyperTraPS. (A)** (i-iv) *Hypercubic transition networks* for different model parameterisations. Here, data are generated from a process where pairs of features influence the acquisition of other features. Inferred transition networks are shown (edge widths give probabilities of transitions) for several models: (i) every edge has an independent parameter; (ii) each feature may have positive or negative influence on the basal rate of each other (as in original HyperTraPS and Mutual Hazard Networks); (iii) each *pair* of features may have additional positive or negative influence on the basal rate of each other feature; (iv) as (i), but following regularisation to remove unimportant edges. (v) *Influence graph* describing influences between features and pairs of features in the $L^3$ model (iii), plotted as in Fig 2C, showing the effect of higher-order interactions. (vi) shows the progress of the regularisation process in (iv), where the initial 384 parameters are progressively pruned (removing model complexity without sacrificing likelihood) until the information criterion (here AIC) starts to increase as further removal of parameters compromises the likelihood. **(B)** *Feature acquisition orderings* from inference of the competing-pathway model for $L = 70$ features, using $2L$ observations, demonstrating the scalability of HyperTraPS-CT. The size of a point reflects the inferred probability that a given feature (vertical axis) is acquired in a given step (horizontal axis) in the accumulation process. The same double-pathway structure as before is used. **(C)** *Hypercubic transition network* from inference of the competing-pathway model using priors to enforce an ordering on the basal rates for each feature: $1 > 5 \gg$ (others). Initial fluxes are reweighted, reflecting the influence of this prior information. **(D)** Using learned hypercubes for predictions. (i) *Predicted features*: predicting the values of hidden features in the observation 1????. Word cloud gives the set of inferred underlying states, with word size corresponding to the probability of that particular state. Bar chart gives the inferred probability that each feature in the underlying state takes the value 1. (ii) *Predicted next step*: Predicting the next step from state 11000. Word cloud and bar chart both reflect the inferred next state to be encountered; the size of the word, and height of the bar, give the corresponding probabilities.

features [4, 30]. Here, we asked whether the algorithm could recover dynamic structure with limited observations of large feature sets. To this end, we increased $L$ in the double-pathway model system above, and took exactly one observation from each transition on the associated network. Fig 3B shows that although convergence takes some time, the continuous-time case can readily identify dynamic patterns with 70 features with only 140 observations (Fig 3B); other large $L$ cases are shown in Fig D in S1 Appendix.

**Maximum likelihood and Bayesian approaches.** As HyperTraPS-CT is concerned with the likelihood estimation for a particular parameterisation, the surrounding inference scheme using this likelihood is flexible. Previous HyperTraPS work was based on a Bayesian picture through Markov chain Monte Carlo (MCMC) [4] or auxiliary pseudo-marginal MCMC (APM MCMC) [24, 43]. We now include approaches for more straightforward likelihood maximisation, including simulated annealing and stochastic gradient descent. These approaches return

a single point estimate for a high-likelihood parameterisation, but typically require substantially less processor time than the Bayesian exploration of parameter space (Fig C in S1 Appendix).

The Bayesian setting allows prior information to be included in the inference process. HyperTraPS-CT currently allows this prior information to be specified in the form of uniform distributions on parameter values, allowing the scale and sign of individual rates and feature interactions to be specified or constrained. A simple example is given in Fig 3C, where a prior is used to constrain the basal rates for each feature in the accumulation process. This has the direct posterior influence of rebalancing the probabilities of the initial steps in the accumulation process, as well as shifting the posterior probabilities of later transitions.

**No, uncertain, or precise timing observations.**   Representation of observation times as time windows (which can have infinite, finite, or zero width) allows different degrees of uncertainty in observation time to be captured. This is of particular use, for example, in phylogenetically embedded data, where the branch lengths connecting an ancestor to a descendant are uncertain; or in cross-sectional snapshot data, where the time since the 'root' state is typically unknown. The effects of increasing or decreasing the uncertainty of observation times in the example system are demonstrated in Figs B and E in S1 Appendix; this ability is exploited in the tuberculosis case study below.

**Predictions of hidden features and future transitions.**   HyperTraPS supports prediction of unseen features (for example, given an inferred transition network, what is most likely to replace the ?s in 001101??0?) and future dynamics (for example, given an inferred transition network, what is most likely to happen next to the state 001101010, and how long will it take?). Simple illustrative examples are given in Fig 3D. These predictions may be useful in applied settings: for example, if a new bacterial strain has some but not all of its drug resistance phenotypes analysed, and clinicians require a prediction of which resistance will evolve next.

**Visualisations.**   As part of the HyperTraPS-CT software implementation, we have included a suite of visualisation procedures allowing interpretation of inference outputs (described in Table 1). These include diagnostic traces of values during the optimisation or MCMC process (Fig C in S1 Appendix); summaries of the inferred dynamics by probabilities of different event orderings (Figs 3B and 4C, Figs B–F in S1 Appendix) and probabilities of different states over time (Figs 2D and 5D); full or truncated visualisations of the inferred transition graph with associated timings (Figs 2A, 3A, 3C, 4A and 5B, Figs E-F in S1 Appendix); and visual reports of predictions from the inferred model (Fig 3D). Depending on the model structure chosen, matrices or graphs of influences between individual features (Figs 2B, 2C, 4B and 5C, Figs F-G in S1 Appendix) or sets of features (Fig 3Av, F in S1 Appendix) can also be produced.

Taken together, these examples demonstrate the capacity of HyperTraPS-CT to work with (i) different model structures (positive and negative pairwise/mutual hazard interactions, influence of larger sets of features on accumulation dynamics, arbitrary logic interactions and completely independent transitions between states); (ii) different data structures (cross-sectional, phylogenetic, longitudinal data with absent, precise, or uncertain observation timings); (iii) different inference approaches (maximum likelihood, Bayesian, different regularisations); (iv) large datasets of many dozen features.

## Cancer progression in acute myeloid leukemia

To demonstrate HyperTraPS' capacity alongside state-of-the-art alternatives, we first look to the cancer progression field, where accumulation modelling is well established [1]. For a comparison with a recent approach, itself compared positively with previous approaches, we use

**Table 1. Visualisations used throughout the article.** The names of each type are given in *italics* in the captions of figures where they appear.

| Visualisation type | Examples in article | Description |
|---|---|---|
| *Hypercubic transition network* | 2A, 3Ai-iv, 3C, 4A, 5B, E and F in S1 Appendix | The inferred transition network for the system. Nodes are states, edges are transitions corresponding to one feature acquisition. Edge thickness gives the probability of each transition in one traversal of the network (transition probability × source state occupancy). Edge labels (sometimes omitted for clarity) give the feature acquired and (where applicable) the time window of the transition. |
| *Influence matrix* (for $L^2$ model) | 2B, 4B, F in S1 Appendix | The bare rates of acquisition of each feature (on-diagonal elements), and how the presence of each other feature influences this base rate (off-diagonal elements). Element $i, j$ for $i \neq j$ describes how the base rate of acquisition of $i$ is influenced by the presence of feature $j$. |
| *Influence graph* (for $L^2$ or $L^3$ models) | 2C, 3Av, 5C, F and G in S1 Appendix | How the presence of individual features (for both $L^2$ and $L^3$ models) or pairs of features (for the $L^3$ model) influence the base rate of acquisition of other features. An arrow from $A$ to $B$ denotes the influence that the presence of $A$ has on the acquisition of $B$. For the $L^2$ case, this provides the same information as the off-diagonal elements of the influence matrix above. |
| *State probabilities with time* | 2D, 5D | The probability that the system is in a given state at a given time. Probabilities are the height of each rectangular region ('motif') corresponding to the labelled state |
| *Feature acquisition orderings* | 3B, 4C, B-F in S1 Appendix | Various visualisations of the ordering in which features are acquired. 'Bubble' plots give the probability (circle size) that feature $i$ is acquired at time $j$. The motif-style plots report the same information using the height of rectangular regions ('motifs'). Histogram plots show the probability of feature acquisition as a function of time. The time series version visualises the (continuous) times at which each feature is acquired as the system accumulates more (progressing up the vertical axis). |
| *Predicted features* | 3Di | Predicted values of hidden features, given an inferred model. A word cloud gives the set of inferred underlying states, with word size corresponding to the probability of that particular state. A bar chart gives the inferred probability that each feature in the underlying state takes the value 1. |
| *Predicted next step* | 3Dii | Predicted next step from a given state, given an inferred model. The size of words in a word cloud, and heights of the corresponding bars, give the probabilities of each state. |

the single-cell genomic dataset of acute myeloid leukemia evolution from [44], previously analysed with TreeMHN [12]. This dataset consists of a set of trees linking 'ancestral' and 'descendant' observations, represented as barcodes describing the presence/absence of a mutation in a set of genes. Following ancestral state reconstruction on these trees, assuming a mutation-free precursor root state, we used HyperTraPS-CT with penalised likelihood to infer the pathways of mutation acquisition during cancer progression. In the absence of explicit timing information in this dataset, we exploit the flexibility of our time window approach and simply assume that transitions occur at some time between an infinitely distant previous horizon and the moment of observation. Fig 4A shows a truncated set of most probable early accumulation steps; Fig 4B shows an inferred map of interactions between mutations; Fig 4C shows the probability that a given mutation occurs at a given ordinal step through the accumulation process.

HyperTraPS-CT with penalised likelihood identifies many of the same features as TreeMHN. The ordering of base rates of mutational changes is very comparable: *DNMT3A* has the highest base rate, followed by *IDH2*, *FLT3*, *NRAS*, and *NPM1*, with the same collection of genes following these. Sets of interactions between changes are also consistently identified, with, for example, *DNMT3A* typically acting to promote the acquisition of other mutations (except *WT1*); *IDH2* having a more limited set of promotion partners; *FLT3* acting to both promote some mutations (*NPM1*, *WT1*, *RUNX1*) and suppress others (*NRAS*); many mutations acting to promote the acquisition of *NPM1* (except for suppressors *RUNX1* and *ASXL1*), and more features. The relative magnitudes of the inferred interactions are also largely consistent between the two approaches, with, for example, the influence of *FLT3* on *NRAS* being the strongest inferred suppression, and the influences of *IDH2* and *ASXL1* on *SRSF2* among the strongest inferred promotions.

The agreement with TreeMHN here (and behaviour in the simple case studies above) is positive support that HyperTraPS-CT provides a consistent way of estimating progressive

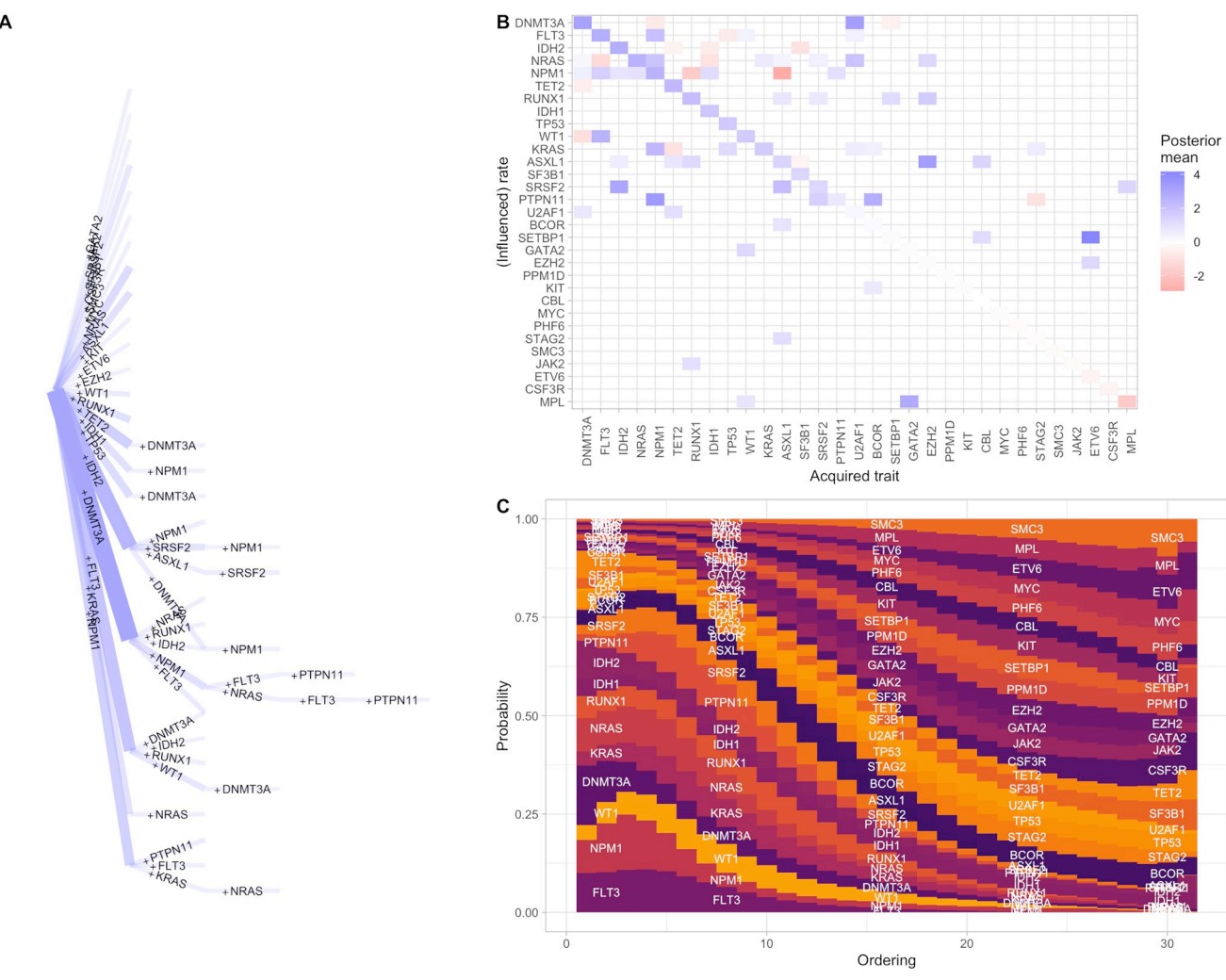

**Fig 4. Accumulation of mutations driving cancer progression. (A)** *Hypercubic transition network* showing the truncated set of inferred high-probability transitions between states in the cancer progression system. Transitions are labelled by acquired feature; only transitions with a posterior probability over 0.006 are plotted. The system proceeds from the leftmost point (no features acquired) and each step to the right corresponds to the acquisition of one feature. Edge widths give probabilities for each transition. **(B)** *Influence matrix* showing of base acquisition rates and positive and negative pairwise influences between features (corresponding network plot in Fig G in S1 Appendix). As in Fig 2B, diagonal elements give base rates, and off-diagonal elements give the influence of an acquired feature (horizontal axis) on the acquisition rate of another feature (vertical axis). **(C)** *Feature acquisition orderings* using a 'motif' plot showing the probability (height of a given rectangular region) that a particular mutation is acquired at a given ordinal step (horizontal axis) in an accumulation process.

accumulation dynamics with real as well as synthetic data. We next looked to a system where its ability to handle uncertain continuous timing data could be demonstrated.

## Acquisition of antimicrobial resistance genes in tuberculosis

To this end, we next used HyperTraPS-CT to explore an evolutionary question of pronounced global health importance—the evolution of antimicrobial resistance—in *Mycobacterium tuberculosis*, using a Russian study [45]. We used the curated data used in [24] where a set of bacterial isolates, related via a known phylogeny, have barcodes corresponding to the presence/absence of resistance to each of ten drugs (Fig 5A). These drugs are referred to by three-letter codes: INH (isoniazid); RIF (rifampicin, rifampin in the United States); PZA (pyrazinamide);

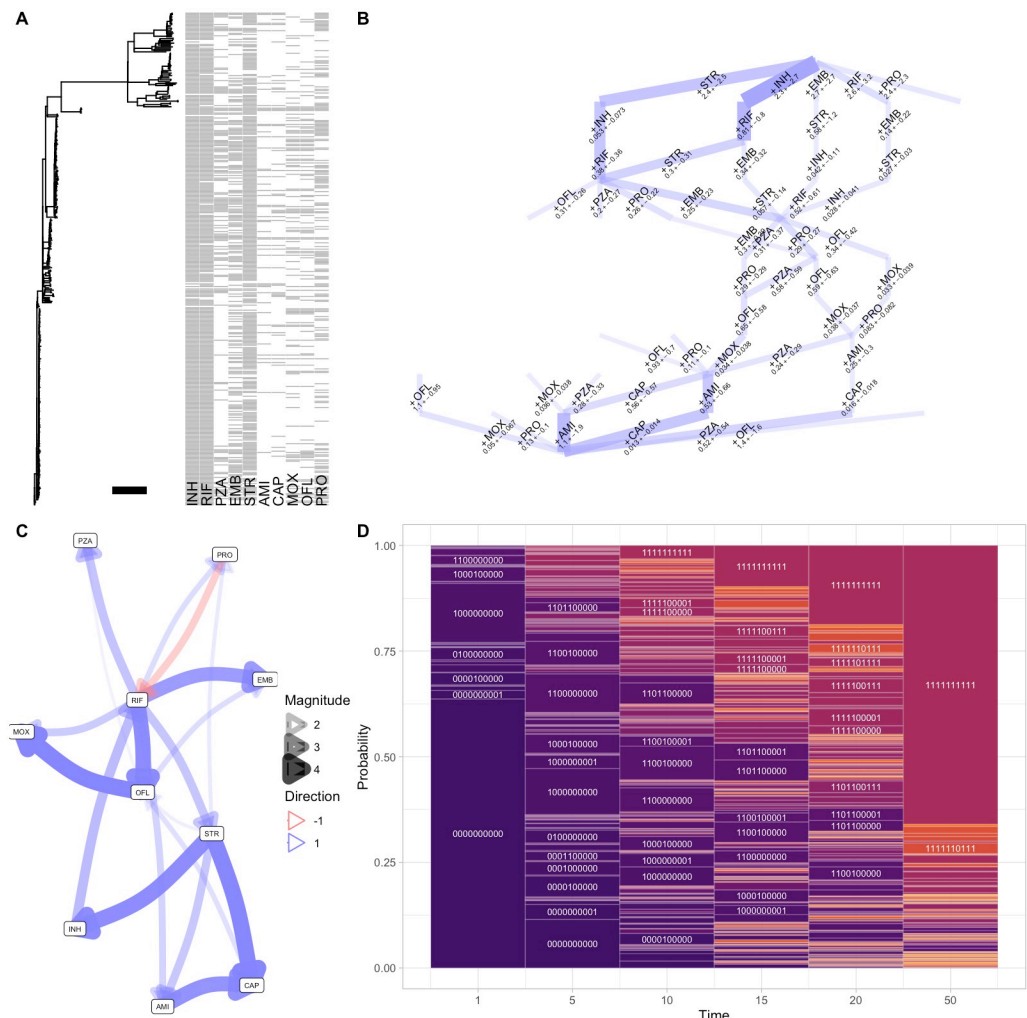

**Fig 5. Inference of the dynamics of anti-microbial resistance evolution in tuberculosis. (A)** Dataset from [45], consisting of a phylogeny connecting observations of drug resistance presence/absence (grey and white pixels respectively). The horizontal scale bar gives a genetic distance of $d = 0.01$ in the original data, interpreted as a value of $\Delta t = 10^3 d = 10$ in the time ordinate for this analysis. **(B)** *Hypercubic transition network* giving inferred high-probability transitions between states in the TB AMR system. Transitions are labelled by the drug to which resistance is acquired in that step, and inferred likely mutational 'timescale'. Edge width gives the probability of a transition; only transitions above a threshold probability of 0.05 are plotted. **(C)** *Influence graph* from the $L^2$ (mutual hazards) picture, describing inferred positive and negative influences (directions 1 (blue) and -1 (red) respectively) between resistance acquisitions. Only influences with a posterior coefficient of variation under 0.3 are plotted. **(D)** *State probabilities with time* as a motif plot describing probabilities of different observed states (heights of corresponding rectangular regions) in the inferred evolutionary dynamics. Further outputs of the inference process are shown in Fig F in S1 Appendix.

EMB (ethambutol); STR (streptomycin); AMI (amikacin); CAP (capreomycin); MOX (moxifloxacin); OFL (ofloxacin); and PRO (prothionamide). We used HyperTraPS-CT with penalised likelihood to learn the pathways of accumulating multi-drug resistance. We reconstructed ancestral states on the phylogeny using the maximum parsimony approach, assuming irreversible accumulation dynamics. The branch lengths on the phylogeny (Fig 5A) do not correspond to absolute timings but to a measure of evolutionary change, estimated from independent genomic data [24, 45]. We first assume that this estimated phylogeny is precise, and use a continuous time inference picture precisely specifying observation time as the branch length $b$ for

each transition ($\tau_1 = \tau_2 = b$ in Eq 1). The time ordinate here is then intepretable as genetic distance between nodes on the phylogeny, scaled by $10^3$ for clarity. The inference results are shown in Fig 5.

Beginning with the inferred accumulation pathways on the hypercubic transition network (Fig 5B), resistance to INH, then RIF/STR in either order, then likely EMB is the dominant pathway. This highest-probability pathway matches observations from discrete-time inference in [24] and [26] (also shown in Fig F in S1 Appendix). However, accounting for the continuous 'time' picture suggests another mode which is less prominent in the discrete-time picture: the early acquisition of STR resistance, followed by INH, then RIF. This pathway emerges in the continuous time picture because, although STR resistance is less common than INH or RIF resistance in the dataset, the transitions involving early STR resistance occur on a shorter time-scale. This information is of course excluded from discrete-time pictures.

Following these early steps, pathways become more branched, with a collection of competing routes and corresponding timescales visible in Fig 5B and 5D. Generally PRO and PZA resistances are likely acquired at intermediate stages, and MOX, OFL, CAP, AMI at later stages. The modal final resistance acquisition sequence is PZA-CAP-AMI. The general ordering of these resistances agrees with [24], but the relatively strong support for AMI (not CAP) as the final resistance step again emerges because of the timings of the associated observations. The map of interactions (Fig 5C) gives a collection of positive interactions between features, and one negative interaction (PRO repressing RIF acquisition).

What if we cannot assume the phylogeny linking these observations is precisely known? If branch lengths are uncertain, the associated observation times for an ancestor-descendant transition are also subject to uncertainty. To demonstrate how HyperTraPS-CT can allow for such uncertainty, we set $\tau_1 = 0$, $\tau_2 = b$ instead of $\tau_1 = \tau_2 = b$ above. In this way, the observation of the descendant state at some time between 0 and $b$ is required, rather than precisely at $b$. The corresponding outputs are shown in Fig F in S1 Appendix. The same dynamics are recovered but with substantially increased uncertainty on transition timescales and fewer robustly supported interactions, reflecting the additional uncertainty in the system.

We also used the model flexibility in HyperTraPS-CT to explore evidence for feature interactions beyong the pairwise cases identified in Fig 5. To this end, we used the $L^3$ model (allowing nonadditive influence of feature pairs on other feature acquisitions) with penalised likelihood on the same tuberculosis dataset. Fig F in S1 Appendix shows that some nonadditive influences of feature pairs on acquisition rates were identified with confidence: for example, a combination of RIF and OFL positively influences the rate of acquisition of MOX.

### Prediction of unobserved and future behaviours in accumulation dynamics

The inferred dynamics here can be used, for example, to provide predictions about the likely next drug resistance to evolve from a given observed state (as in Fig 3Dii), and to predict phenotypic features from an imperfectly sampled new observation (as in Fig 3Di). Both of these predictive statements could be of conceivable applied use. A prediction that resistance to a given drug is likely to occur next might suggest the use of a different drug (one with lower and/or later acquisition propensity). In cases where phenotypic assays of drug resistance are challenging (requiring researcher time and resources), the predictions of unmeasured drug resistances could be used as a (clearly imperfect) substitute.

To test the capacity of HyperTraPS-CT to make such predictions, we split the transition set for the tuberculosis case study into 75:25 training:test subsets, and used the training subset to

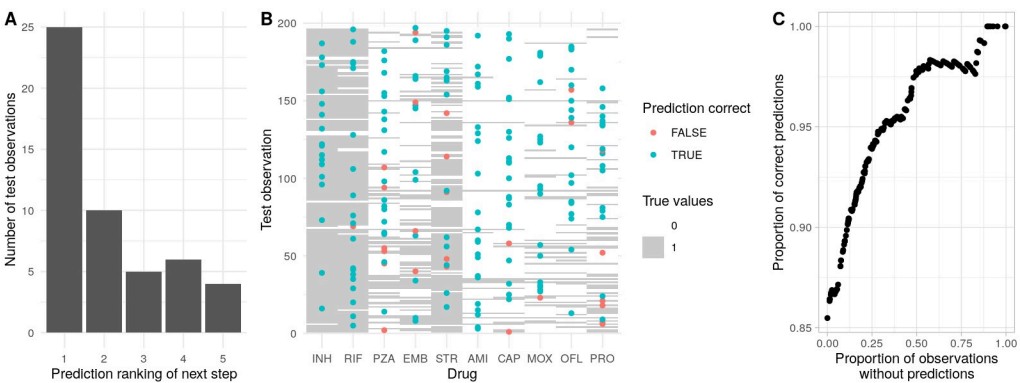

**Fig 6. Predictions of future and unobserved features in tuberculosis multi-drug resistance evolution. (A)** Following training, predictions were made for the next accumulation step in a set of withheld test observations (see text). For each observation, each possible step from the precursor state was ranked by predicted probability. The plot shows the predicted ranking of the true acquisition (1 predicted to be most likely) across test observations. **(B)** Prediction of artificially obscured features in test data. The 'barcode' background shows the full test data, the points correspond to an artificially obscured feature that was predicted from the trained model, coloured by whether the prediction was correct or not. 86% of predictions were correct in this example. **(C)** More generally, a parameterised choice can be made about whether to assign a confident prediction based on its posterior probability. This plot shows how the proportion of correct predictions increases as this 'strictness' criterion is increased, at the cost of a higher proportion of observations without an assigned prediction.

produce posteriors on the hypercubic model. We did this 75:25 split ten independent times to guard against accidents of sampling: the following results are amalgamated across these instances. To address the first predictive task, using the unseen test data, we identified transitions where the 'ancestral' and 'descendant' states differed by one acquired feature, and asked whether acquisition of that inferred feature was among the most probable predicted steps from the ancestral state. In the majority of cases, the true acquisition was the most or next-most likely step predicted by the trained model, with the most-likely predicted step being clearly most common (Fig 6A).

We next artificially obscured a random set of features in the test data and attempted to predict their values based on the remaining features and the model inferred from the training data (Fig 6B). For each obscured feature $F$ we recorded $P(F = 1)$, the posterior probability that it would take value 1, from the inferred model. With a 'strictness' parameter $\Delta$, we assigned a prediction of 1 for features with $P(F = 1) > 0.5 + \Delta$, a prediction of 0 for $P(F = 1) < 0.5 - \Delta$, and no prediction otherwise. As $\Delta$ changes from 0 to 0.5 we thus require stronger posterior evidence for making a prediction. Fig 6B shows the case for $\Delta = 0$, where 86% of predictions were correct; the general behaviour of prediction accuracy against proportion of assigned predictions is shown in Fig 6C. Enforcing stricter criteria for assigning a prediction readily amplifies accuracy over 95% at the cost of roughly a quarter fewer confident predictions.

Taken together, HyperTraPS-CT has demonstrated both compatibility with existing approaches for the study of cancer progression and in synthetic case studies, and the ability to exploit timing information to refine estimates of evolutionary dynamics in the study of antimicrobial resistance. At the same time, its flexibility will allow a range of scientific questions to be explored across other applied fields. These include the role of higher-order interactions between features in determining accumulation behaviour, and (with approaches to compare inferred transition graphs [28]) the similarities, differences, and modulating factors of accumulation dynamics in different samples.

## Discussion

We have introduced a continuous timescale to hypercubic transition path sampling (Hyper-TraPS), a flexible approach to inferring the dynamics of accumulaton models. We have also generalised the parameter structures used in the underlying model so that arbitrarily high orders of interaction between feature sets can be captured by the inference process, and introduced a panel of new options for predictions, inclusion of prior knowledge, optimisation approaches, regularisation, and visualisation. Higher-order interactions have previously been shown to have explanatory power in, for example, ovarian cancer progression [26], and limited higher-order interactions between drug resistances are observed here in the tuberculosis case study. We hope that the regularisation and model selection approaches we provide here will help explore potentially useful model structures in other contexts.

A question of causality arises in considering the inferred interactions and influences between features. In several biological settings, some features may be expected to act as 'drivers' (actively influencing others) and some as 'passengers' (subject to, but not exerting, influence). Mutations in cancer progression are a clear example [46]. In the network of influences between features inferred with accumulaiton modelling, one would expect 'passengers' to be identified with few outgoing edges (no influences on other features) and 'drivers' to be identified with more outgoing edges (influencing several other features). But population-level effects and combinations of different fitness influences from different features can complicate this identification, making it important to consider inferred influences through the lens of a particular evolutionary model for the process under study [2].

We have shown that the consideration of continuous timing information can lead to differences in the outcome of inference—as demonstrated by the ordering of streptomycin (STR) resistance in the tuberculosis case study. Features that are less represented in a dataset, but which are associated with rapid acquisition times when they do appear, will tend to be assigned later acquisition orderings in explicitly or implicitly discrete-time approaches. Timing information can help resolve these dynamics—but is typically uncertain, as feature acquisitions can occur anywhere between two sampling events. We have outlined and demonstrated one approach by which such uncertainty may be addressed—through including a time window of controlled length during which an observation may be made. This approach corresponds to a particular error model—a uniform distribution over possible observation times. Generalising this model to include different distributions over observation times constitutes a target for future work. The robustness of our approach can also be tested by artificially perturbing the source dataset and assessing the influence of these perturbations on the posteriors [4].

As with other approaches, the posteriors that HyperTraPS initially produces reflect an 'umbrella' picture of accumulation dynamics, combining evidence from observations that may have been subjected to different selective pressures and environmental conditions [4, 24]. This umbrella picture reflects coarse-grained posterior knowledge of accumulation pathways given all observations—in a sense, describing the distribution of belief about a putative new sample about which we have no further information. Refinement of these umbrella posteriors to a finer-grained picture is possible by including information about the context of different samples. Previous examples include the broader taxonomy of an individual in evolutionary dynamics [5], level of patient risk in disease progression dynamics [9], and an individual's ecology in behavioural dynamics [13]. In such cases, observed multimodality in the umbrella posteriors can sometimes be accounted for (at least partly) by a natural distinction between data subsets from differently categorised observations, informing about mechanistic influences on the accumulation process. Heterogeneity amongst the entities of study is also a consideration for these accumulation models. For example, tumours are internally heterogeneous, and

individual cells within a sampled tumour may have different mutational profiles [34, 35, 47, 48]. As discussed in [2], these methods can still be applied to bulk samples of these heterogeneous cases (although single-cell observations are clearly more powerful), but the interpretation must be altered accordingly—inferences now apply not to within-cell dynamics, but to the observed mixture of states.

In some contexts, the properties of some observed states may not be completely known. Uncertainty in descendant observations (and hence anywhere in cross-sectional experimental design) does not challenge our approach, which readily supports a 'missing data' flag in any number of features for such observations. However, uncertainty in ancestral states in the longitudinal or phylogenetic context poses a larger technical challenge [4]. Our approach can always be applied using a method originally called 'phenotypic landscape inference': uniform, unbiased sampling with likelihoods estimated by tracking the number of states compatible with uncertain observations (essentially using brute force sampling to estimate the left-hand side of Eq 1), as originally implemented in [5]. However, the question of how to *most efficiently* sample evolutionary paths from an uncertain origins will be addressed in future research.

In case where ancestral states are not directly observed, the question of how to reconstruct evolutionary transitions arises. As described in Methods, for simple evolutionary dynamics a parsimony-like assumption can be used, where ancestors are assumed to possess a feature if and only if both descendants possess it. Parallel acquisitions (or losses) of features in sister lineages are thus undercounted. Camin-Sokal [40] or Wagner [41, 42] methods for reconstructing ancestral states could be used, but for large problems could give rise to a large set of putative transition sets to consider, dramatically increasing the complexity of the problem. A simpler remedy for this undercounting would be to assume the (likely incorrect) opposite: that each feature present in each observation reflects an independent acquisition (in other words, no features are inherited from ancestors). Any inferences that are consistent across these two different pictures can then be regarded as robust with respect to either under- or over-counting of parallel acquisitions. However, in many cases, the initial parsimony picture is compatible with the biological context of the evolutionary process.

HyperTraPS by its nature is a sampling approach, and does not have the native capacity for exact calculations of transition path probabilities as found in other approaches including MHN [33] and HyperHMM [26]. For small systems, the likelihoods estimated from this sampling approach are effectively indistinguishable from the exact results (Fig B in S1 Appendix), and the precision of this estimation can be controlled (at the expense of computational time) by the number of sampled paths $N_h$. The inevitable expansion of pathway and parameter space as the number of features $L$ increases means that sampling approaches are currently used even when using MHN for large systems [12]. However, the necessarily random nature of the sampling underlying HyperTraPS must be considered in its responsible interpretation; convergence of results from different random number seeds, for example, should be assessed as a 'safety check'. On a related note, cases will clearly exist in accumulation modelling where different parameterisations may have equal abilities to reproduce observations. The Bayesian embedding of HyperTraPS, and the approaches for regularisation and model comparison we suggest here, can be used in such cases where identifiability is challenged, to allow reporting of the flexibility and constraints on different mechanistic parameters under different model structures.

As approaches for accumulation modelling, traditionally grounded in cancer statistics, gain more and more features in common with evolutionary biology methods (including connections with trees/phylogenies, and continuous time), it is worth reiterating that methods for discrete Markov dynamics on trees have existed in the evolutionary literature for decades [49–51]. The Mk (Markov k-state) model is well established in evolutionary biology and any

discrete Markovian model on a phylogeny (including a 'star' phylogeny, corresponding to independent instances / cross-sectional observations) can in principle be viewed as a subset of this picture [16]. Recent work has exploited this connection to allow inference of reversible accumulation dynamics, albeit for small numbers of features [52]. In accumulation modelling the focus is typically on features and their relationships, rather than states and their relationships, and the different dependency structures (pairwise interactions, logic interactions, and so on) are often explicitly encoded in accumulation models in a way that would be less straightforward to extract—and potentially impossible to infer in reasonable computational time—from an Mk model picture. But the connections between the disciplines will be worth exploring in future developments.

## Supporting information

**S1 Appendix. Mathematical derivations, validation, development, and additional results for HyperTraPS-CT.**
(PDF)

## Acknowledgments

The authors are grateful to Ellen Røyrvik, Paul Una, Amelia Earl, and Ai Monti for valuable discussions.

## Author Contributions

**Conceptualization:** Olav N. L. Aga, Morten Brun, Ramon Diaz-Uriarte, Konstantinos Giannakis, Iain G. Johnston.

**Data curation:** Iain G. Johnston.

**Formal analysis:** Iain G. Johnston.

**Funding acquisition:** Ramon Diaz-Uriarte, Iain G. Johnston.

**Investigation:** Olav N. L. Aga, Ramon Diaz-Uriarte, Iain G. Johnston.

**Methodology:** Morten Brun, Kazeem A. Dauda, Ramon Diaz-Uriarte, Konstantinos Giannakis, Iain G. Johnston.

**Project administration:** Iain G. Johnston.

**Software:** Olav N. L. Aga, Ramon Diaz-Uriarte, Konstantinos Giannakis, Iain G. Johnston.

**Supervision:** Iain G. Johnston.

**Validation:** Ramon Diaz-Uriarte, Iain G. Johnston.

**Visualization:** Iain G. Johnston.

**Writing – original draft:** Iain G. Johnston.

**Writing – review & editing:** Olav N. L. Aga, Ramon Diaz-Uriarte, Konstantinos Giannakis, Iain G. Johnston.

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
