## [Decision Letter · Decision Letter 0]

8 Jun 2024

Dear Dr Johnston,

Thank you very much for submitting your manuscript "HyperTraPS-CT: Inference and prediction for accumulation pathways with flexible data and model structures" for consideration at PLOS Computational Biology. As with all papers reviewed by the journal, your manuscript was reviewed by members of the editorial board and by several independent reviewers. The reviewers appreciated the attention to an important topic. Based on the reviews, we are likely to accept this manuscript for publication, providing that you modify the manuscript according to the review recommendations.

Sincerely,

Mark Alber, Ph.D.

Section Editor

PLOS Computational Biology

Mark Alber

Section Editor

PLOS Computational Biology

Reviewer's Responses to Questions

**Comments to the Authors:**

Reviewer #1: The paper reports an extension to a sequence of results and tools the

authors have worked on in the past years. The *HyperTraPS-CT* tool

extends the sampling methods the authors have worked on, by adding the

capability to include (continuous) wall time in its core algorithms.

The added capabilities allow HyperTraPS-CT to expand its posterior

probability constructions to a wide range of phenomena and systems

where a temporal model reconstruction is needed. In particular the

authors name *cross-sectional*, *phylogenetic* and *longitudinal* data

where various combinations of features can be analyzed (combination

can be ternary and beyond).

The paper concludes by showing the application of HyperTraPS-CT to two

recent and varied datasets, one the well-known Morita's SCs AML

dataset, used to infer cancer progression, and a second one to check

acquired resistance in tuberculosis. The results reported in the

applications do make sense and are useful as supporting arguments

about the goodness of the approach.

The paper is very well written and full of well presented

information. Each choice presented is properly contextualized and

explained. The resulting tool, HyperTraPS-CT appears to be

significant improvement over the state of the art, even considering

the previous non-CT version. I have not seen paper of this quality and

usefulness in a while.

Nevertheless, I do have a few suggestions for improvement.

Although this is implied in much of the paper and of the state of the

art, it would be useful to clearly state the overall computational

complexity of the algorithms making up HyperTraPS-CT.

On a related point, I would suggest the authors to include a separate

"box" explaining the different sampling strategies mentioned in the

state of the art. This would make the paper more self contained.

A separate and expanded section or box, much closer to the beginning

of the "Methods" section about the "visualization" tools is, IMHO,

opinion necessary. Without it, interpreting the figures takes quite

an effort.

Most figures could be split in two, with more explanation in the

captions, especially when the visualization requires much

interpretation.

Finally, I would suggest to make the tool available on BioConductor;

it is already an R application after all.

Reviewer #2: The authors proposed an extension of HyperTraPS (Hypercubic Transition Path Sampling) to the case of continuous time dynamics. The original HyperTraPS method is a Bayesian inference framework which models the accumulation of features in disease progression in evolutionary biology with a discrete time Markov chan, where the space of states is visualised as an hypercube in which vertices are the accessible states of the system and edges the transition steps between them. In their extension to continuous the processes, the authors provide a semi-analytical approach to compute the transition probabilities between the states of the hypercube. The transition between states are assumed to be Poissonian and the waiting time distribution of a single trajectory ais modelled as an hyperexponential distribution, while the marginalization integral summing over all possible trajectories connecting different states are approximated by path sampling on the hypercube. Given the tipically large number of states, in order to reduce the space of allowed trajectories, the authors introduced what they called “compatibility condition”, meaning that in a Markov chain step the system can acquire at most a feature in accumulation processes, or lose at most a feature in loss processes.

The authors proposed different type of parametrizations of the rates of the Markov chain which allow to capture higher order interactions between features and different regularization schemes as recursive feature elimination or additional terms in the likelihood penalizing model complexity during the learning process, which relies on a MCMC framework. The method is tested by the authors on simulated data increasing number of features and proposed two case of study in cancer progression of acute myeloid leukaemia and antimicrobial resistance genes acquisition in tubercolosis. The authors exploited these datasets to test the performance of the tools in predicting future and unobserved prediction in accumulation dynamics.

I appreciated the effort of the authors to extend HyperTraPS and establish a continuous time framework to study disease progression. In particular, I found very interesting the improvements regarding rates parametrization, the proposal of different regularisation schemes and the posterior estimates of disease pathways and feature correlations. I have some questions regarding the use of the method in certain contexts that may be interesting in cancer evolution:

1) The concept of time is never clarified through the paper. The authors mentioned two single cell datasets for which they have ancestral relations between the genomic variants. Do they assign to any variant a pseudo-time related to its occurrence location in the cell tree? Do they assign to group of variants a time window of arrival identified by the times of barcode insertions? I think the authors may clarify which time scale they consider in their case of studies.

2) The authors assumed a compatibility condition in order to reduce the space of allowed trajectories of the system. This implies that the system can only acquire variants in accumulation processes, one per each step of the trajectories, and can never lose them. However, this may not be true, since evolution is always branched ([1]), and two populations collected at different time points will have a common ancestor and a set of common variants, the ones inherited by the common ancestor, and possibly a subset of private variants that appear only in one of the two samples. So, in general it is not true that the genomic state of system at the second time point has the genomic state at the first time point as boundary condition. Conversely, both the states should be considered as independent evolution of the system from a common boundary condition given by their common ancestor. Is it possible to account for such aspect of longitudinal data in the present framework?

3) The authors implicitly assume that the population has no sub clonal structure and a system can be identified with a set of variants, which are therefore clonal. This may be not true in general [1] and it has been observed that a tumour may have polyclonal structure at detection time. In such scenario, the state of the system should be thought as a mixture of states, with some variants present only in a subset of cells. Can the present framework account for polyclonal states of the cell population?

4) Is it possible from the estimated feature correlations and predicted pathways to distinguish between drivers and passenger variants? The former are intended as genomic alterations that drive the evolution, providing fitness advantage or resistance, while the latter are variants that do not confer any fitness advantage [1]. The passenger variants that arrive on the genome of a cell before this acquires a driver raise their frequency [3] and in case of clonal sweep are found to be clonal at sample collection, even if they do not have any role in driving the disease progression. I think it would very interesting if it was possible to predict from the Markov chain dynamics the role of the variants and in particular which ones have a driving roles in the different trajectories of the disease progression and which ones are simply passengers.

References

[1] Turajlic, S., Sottoriva, A., Graham, T. et al. Resolving genetic heterogeneity in cancer. Nat Rev Genet 20, 404–416 (2019). https://doi.org/10.1038/s41576-019-0114-6

[2] Williams, M.J., Werner, B., Heide, T. et al. Quantification of subclonal selection in cancer from bulk sequencing data. Nat Genet 50, 895–903 (2018). https://doi.org/10.1038/s41588-018-0128-6

**Have the authors made all data and (if applicable) computational code underlying the findings in their manuscript fully available?**

Reviewer #1: Yes

Reviewer #2: Yes

PLOS authors have the option to publish the peer review history of their article (what does this mean?). If published, this will include your full peer review and any attached files.

Reviewer #1: No

Reviewer #2: No

Figure Files:

Data Requirements:

Reproducibility:

References:

---

## [Decision Letter · Decision Letter 1]

6 Aug 2024

Dear Dr Johnston,

We are pleased to inform you that your manuscript 'HyperTraPS-CT: Inference and prediction for accumulation pathways with flexible data and model structures' has been provisionally accepted for publication in PLOS Computational Biology.

Best regards,

Alison Marsden

Section Editor

PLOS Computational Biology

Mark Alber

Section Editor

PLOS Computational Biology

Reviewer's Responses to Questions

**Comments to the Authors:**

Reviewer #1: The authors responded to most requests I made and I am satisfied with them. The paper is now more readable and

each piece of information better qualified. Yet, I still would have broken up some of the pictures.

Minor issues.

In Figure 3 there is a missing reference: "(D) Using learned hypercubes for predictions. (i) Predicted features: predicting the values of hidden features in the observation 1????.

I could not access the current Supplementary Material Figures.

Reviewer #2: The authors provided exhaustive answers to the questions i have made in the previous review. They included several examples and references and added the discussion points in the main text.

**Have the authors made all data and (if applicable) computational code underlying the findings in their manuscript fully available?**

Reviewer #1: Yes

Reviewer #2: Yes

PLOS authors have the option to publish the peer review history of their article (what does this mean?). If published, this will include your full peer review and any attached files.

Reviewer #1: No

Reviewer #2: No

---

## [Editor Report · Acceptance letter]

26 Aug 2024

PCOMPBIOL-D-24-00405R1 

HyperTraPS-CT: Inference and prediction for accumulation pathways with flexible data and model structures

Dear Dr Johnston,

I am pleased to inform you that your manuscript has been formally accepted for publication in PLOS Computational Biology. Your manuscript is now with our production department and you will be notified of the publication date in due course.

With kind regards,

Olena Szabo
